# Place-specific factors associated with adverse maternal and perinatal outcomes in Southern Mozambique: a retrospective cohort study

Prestige Tatenda Makanga,[1] Charfudin Sacoor, Nadine Schuurman,[2] Tang Lee,[3] Faustino Carlos Vilanculo,[4] Khatia Munguambe,[4] Helena Boene,[4] Ugochinyere Vivian Ukah,[5] Marianne Vidler,[3] Laura A Magee,[6] Esperanca Sevene,[4,7] Peter von Dadelszen,[6] Tabassum Firoz,[8] On behalf of the CLIP Working Group

For numbered affiliations see end of article.

**Correspondence to**
Dr Prestige Tatenda Makanga; makangap@staff.msu.ac.zw

## ABSTRACT

**Objectives** To identify and measure the place-specific determinants that are associated with adverse maternal and perinatal outcomes in the southern region of Mozambique.

**Design** Retrospective cohort study. Choice of variables informed by literature and Delphi consensus.

**Setting** Study conducted during the baseline phase of a community level intervention for pre-eclampsia that was led by community health workers.

**Participants** A household census identified 50 493 households that were home to 80 483 women of reproductive age (age 12–49 years). Of these women, 14 617 had been pregnant in the 12 months prior to the census, of which 9172 (61.6%) had completed their pregnancies.

**Primary and secondary outcome measures** A combined fetal, maternal and neonatal outcome was calculated for all women with completed pregnancies.

**Results** A total of six variables were statistically significant (p≤0.05) in explaining the combined outcome. These included: geographic isolation, flood proneness, access to an improved latrine, average age of reproductive age woman, family support and fertility rates. The performance of the ordinary least squares model was an adjusted $R^2=0.69$. Three of the variables (isolation, latrine score and family support) showed significant geographic variability in their effect on rates of adverse outcome. Accounting for this modest non-stationary effect through geographically weighted regression increased the adjusted $R^2$ to 0.71.

**Conclusions** The community exploration was successful in identifying context-specific determinants of maternal health. The results highlight the need for designing targeted interventions that address the place-specific social determinants of maternal health in the study area. The geographic process of identifying and measuring these determinants, therefore, has implications for multisectoral collaboration.

**Trial registration number** NCT01911494.

## Strengths and limitations of the study

► This study's methods draw from the value to mixing methods and validates the importance of variables through first seeking community perspectives, followed by expert opinion through a Delphi consensus and lastly a rigorous and thorough process of selecting the most important statistical variables.

► Small area analyses are implemented to granularise the data and reveal patterns that are normally masked in national averages to elucidate more locally specified linkages between health outcomes and associated determinants.

► The key statistical association that have been uncovered in this research, particularly those from the geographically weighted regression, are specific to this region of Mozambique and may need to be validated before generalising to other settings. Some of the variables that are significantly related with the combined outcome (eg, latrine score) may be a symptom of a deeper socioeconomic problem that this research did not have data and evidence to elucidate.

► All maps were created using data generated from the study. Methods of creating the neighbourhood boundaries have been published in a separate manuscript.

## INTRODUCTION

Improving maternal health has been a global health priority for the past four decades. As global attention shifts from surviving pregnancy and childbirth to ensuring that women thrive throughout their life course,[1 2] much remains to be done to lessen the harmful consequences of pregnancy and childbirth. Maternal deaths are only a small portion of the global maternal burden of ill health; it is estimated that for each death, nearly 20 additional women suffer from life-long disabilities as a result of severe pregnancy-related morbidity.[3 4] In line with that, Sustainable Development Goal 3 aims to improve maternal care and focuses on life-long

BMJ

well-being for women and children, particularly in low-income and middle-income countries where the maternal mortality rate is up to 14 times more prevalent than in high-income countries.[5]

The known risk factors leading to maternal and perinatal death and morbidity exist both at the level of the individual women and at the level of her community. Some of these factors, although experienced at the individual level, are a function (at least in part) of the broader sociocultural environment, making it hard to separate the two. Individual-level factors include maternal age,[6] level of education for women and their partners,[7] contraceptive use,[8] birth spacing,[9] marital status, social standing, self-esteem and psychosocial stress.[10] Environmental and community-level factors that elevate the risk for adverse pregnancy outcomes include physical isolation from health facilities,[11] living in war zones,[12 13] natural disasters[14] and religion[13 15] among others.

Maternal Mortality Ratio (MMR) estimates in Mozambique vary from approximately 250 to 408 deaths per 100 000 live births[16–18] and is among the top 20 countries with the highest MMRs globally. There has been approximately 50% reduction in MMR from 541/100 000 live births since 1990, largely due to falling rates of maternal death resulting from direct (as opposed to indirect) obstetric causes.[16] In contrast, there has been a relative increase in maternal deaths from indirect causes (such as HIV and malaria) over the same period, as fewer obstetric interventions address these conditions.

Previous studies conducted in Mozambique have documented some of the determinants of maternal health in the country. In Southern Mozambique, women who suffered severe maternal morbidities reported that lack of money for transportation and poor road infrastructure and long distances to health facilities caused delays in reaching health facilities, when they sought emergency pregnancy-related care.[17 19] Flooding can isolate communities from obstetric care for months at a time.[20] A cultural acceptance of the male decision maker's absence from matters concerning the woman's pregnancy may contribute to increasing the vulnerability of pregnant women[21] who may not be empowered to make decisions concerning their pregnancies.[19] Prevailing misconceptions around the causes of pregnancy-related illness[22] are likely to influence women's choices to access care.

Emerging global maternal health strategies acknowledge the broad nature of the associated determinants of health and call for multisectoral approaches to complement health system enablers for improving maternal, fetal and child health along the continuum of care.[1 23] These strategies mirror the aim of the Sustainable Development Goals (SDGs) to 'draw on contributions from indigenous peoples, civil society, the private sector and other stakeholders, in line with national circumstances, policies and priorities'.[2] Further to that, the SDGs call for greater measurement of disaggregate subnational trends in life-course health outcomes and associated determinants.[2] This drive to understand the granular population health trends and associated determinants will likely help to better elucidate the place-specific nature of these associations with maternal and perinatal health outcomes.

The aim of this study was to identify and measure the community specific determinants that are associated with maternal and perinatal ill health in the southern region of Mozambique. In line with the recommendations from the SDGs, the study sought to gain a local understanding of these determinants and how their associations with adverse maternal outcomes varied geographically.

## METHODS
### Study setting
The study was conducted as part of the feasibility study for the Community-Level Interventions in Pre-Eclampsia Trial (CLIP) in 2014 in Mozambique. CLIP was a community-based cluster randomised controlled trial aimed at reducing all-cause maternal and perinatal mortality and morbidity in the study region. CLIP was led by the University of British Columbia, in partnership with the Centro de Investigação em Saúde de Manhiça in Mozambique. The feasibility study for CLIP was conducted in 36 administrative regions termed localities within two provinces in the southern part of the country.

### Study design
There were four core aspects of this project that are summarised in figure 1: (1) gathering data on community perspectives of the determinants of maternal health, (2) prioritising variables through a Delphi consensus, (3) collecting primary empirical data on the variables and (4) conducting spatial and statistical analyses to explore the association of these variables with adverse maternal outcomes.

### Community perspectives on the determinants of maternal health
Ten focus groups discussions (FGDs) were conducted in 4 of the 12 clusters in the CLIP study area: Messano, 3 de Fevereiro, Ilha Josina+Calanga and Chongoene. These FGDs involved pregnant women, women of reproductive age, matrons (local birth attendants), male partners, community leaders and community-based health workers. Using purposive sampling combined with snowball sampling techniques, participants for the FGDs were recruited with the assistance of community gatekeepers. The FGDs covered topics regarding the sociocultural, environmental and economic factors thought to be related to adverse maternal events.

Semistructured interviews were conducted with the chiefs in all 12 administrative posts in the study region to better understand the historical context (eg, civil wars, natural disasters, foreign aid and microfinance) of the communities and how these could impact maternal health.

The FGDs and interviews were conducted in a local language (Changana) following a guide of open-ended questions reflecting the topics, were audio-recorded,

**Figure 1** Design overview.

transcribed and translated verbatim into Portuguese before a final translation into English.

The full details concerning data collection, coding of the data and thematic analysis have been previously published.[24]

## Prioritising variables

A Delphi consensus meeting by teleconference was conducted to prioritise the variables for statistical analysis. The Delphi technique helps with 'achieving convergence of opinion concerning real-world knowledge solicited from experts within certain topic areas'.[25] The panel of 17 experts had a range of relevant backgrounds, including obstetrics (n=2), epidemiology (n=2), demography (n=2), health geography (n=1), environmental health (n=1), spatial statistics (n=1), health equity (n=1), health systems research (n=1), medical anthropology (n=4) and mobile health (n=2). A structured questionnaire that had been designed based on an extensive literature review was used as a guide for the Delphi process during the teleconference. The same questionnaire was sent to members of the Delphi group that could not make the call. Consensus was reached after the first round as many of the variables tabled before the experts were backed by literature.

## Participants and data sources

The context-specific variables identified for consideration of their association with a combined maternal and perinatal adverse outcome were collected through a household census conducted as part of the CLIP feasibility study.[26] The census included information on all women

who had been pregnant in the 12 months prior to the census, as well as women of reproductive age who had died. Data collected included individual-level variables (eg, age, education and pregnancy history), as well as community characteristics (eg, availability of the household head and community support initiatives). All reports of maternal, fetal or perinatal deaths were followed up with verbal autopsy[27] to classify the cause.

The census identified 50 493 households that were home to 80 483 women of reproductive age (age 12–49 years). Of these women, 14 617 had been pregnant in the 12 months prior to the census, of which 9172 (61.6%) had completed their pregnancies. For the mother, there were 18 deaths (204.6 MMR) of which the verbal autopsy identified that 38% were from direct causes and 62% from indirect causes. For the baby, there were 288 (3.0%) miscarriages, 466 (4.9%) stillbirths and 8796 (92.1%) live births, of which there were 117 neonatal deaths. A full description of the health and sociodemographic profile of the women of reproductive age in the study area has been published.[26]

In addition, we collected five geographical variables using geographical information systems. There were three travel times to (1): primary health facilities, (2) secondary health facilities and (3) tertiary health facilities, using mixed transport modes for public transport and ambulances. Walking times to the nearest main road (4) were calculated to measure the degree to which communities were geographically isolated. Finally, an indicator for flood proneness (5) was

**Table 1** Community level variables potentially associated with the rates of adverse maternal outcomes

| Community-level variable | Description *(variables calculated for reproductive age women with completed pregnancies)* |
|---|---|
| **Census variables** | |
| 1. Age of reproductive age woman | Average age of reproductive age woman. |
| 2. Household head's education | Average number of years that household heads (man or woman) have spent in school *(no schooling=0; at least primary=7; at least secondary=12; at least a degree=16; graduate=18, postgrad=20).* |
| 3. Household head's availability | Percentage of households where the household head lives in the house. |
| 4. Water source score | Percentage of households that have an improved water source. |
| 5. Latrine score | Percentage of households that have an improved latrine. |
| 6. Private transportation score | Percentage of reproductive age women who live in a house where someone owns a private car. |
| 7. Reproductive age women's education | Average number of years that reproductive age women have spent in school *(no schooling=0, grade 5=5, grade 7=7, grade 10=10, grade 12=12, bachelors=16, graduate=18, postgrad=20).* |
| 8. Fertility rate | Average number of children born to each woman in the community that had a completed pregnancy. |
| 9. Reproductive age women's marital status score | Percentage of reproductive age women in a marital union (monogamous or polygamous) relative to total with completed pregnancies. |
| 10. Reproductive age women's unemployment rate | Proportion of reproductive age women that do not work compared with total reproductive age women with a completed pregnancy. |
| 11. Family support | Percentage of reproductive age women that would receive financial, transport and emotional help from family or neighbours for a pregnancy-related need. |
| 12. Community group support | Percentage of reproductive age women that would receive financial, transport and emotional help from a community-based group for a pregnancy-related need. |
| 13. Financial autonomy in pregnancy | Percentage households where the reproductive age woman is empowered to make financial decisions concerning her pregnancy. |
| **Geospatial variables** | |
| 14. Access to primary health facilities | Average travel time to the nearest primary health facility, using public transport. |
| 15. Access to secondary health facilities | Average travel time to the nearest secondary health facility, using a mix of public transport and an ambulance. |
| 16. Access to tertiary health facilities | Average travel time to the nearest tertiary health facility, using a mix of public transport and an ambulance. |
| 17. Isolation | Average walking time to the nearest main road. |
| 18. Flood proneness | The difference between the road quality indicator (RoQI) score on a typical day in the dry season and on the worst day in the wet season. RoQI scores range between 0 and 100 and are a function of the quality of roads in a community. |

designed based on flood and precipitation records from the previous year.[28] These variables and other community-level estimates for the variables captured in the census were calculated for each locality in the study area as described in table 1. Both the census and geographical data were aggregated into community-level averages at the locality level for each of the chosen variables, as ethical approval did not allow to analyse the location data at the level of the individual woman.

## Statistical methods and variables

The primary outcome for this study was a combined maternal and perinatal outcome that included maternal, fetal and neonatal deaths. The denominator was the total number of live births. A composite outcome was chosen as powering the study for maternal death alone would have required a prohibitively large sample size. There is clinical plausibility in combining the three outcomes as both fetal and early neonatal outcomes are related to the woman's condition during the antenatal and intrapartum periods, while her environment and socio-cultural circumstances have an impact on late neonatal outcomes.[29 30] Furthermore, a significant proportion of miscarriages are related to placental dysfunction, as are stillbirths and neonatal deaths.[31 32]

The spatial statistics module within ArcGIS software[33] was used for exploratory regression to further prioritise variables and to create the global ordinary least squares

**Table 2** Criteria for variable selection prior to regression modelling

| Criteria | Description | Threshold |
|---|---|---|
| Coefficient p value | The CI required for p values of coefficients. | <0.05 |
| Variance inflation factor | Measures redundancy of multicollinearity between the explanatory variables. | <7.5 |
| Jarque Bera p value | Measures whether the model residuals are normally distributed. | >0.1 |
| Spatial autocorrelation p value. | Check for spatial clustering of model residuals. | >0.1 |

(OLS) regression model. The exploratory regression exercise evaluated different combinations of our explanatory variables for their fit for an OLS model and how these explained trends in our outcome variable. This method implements the exploration by screening variables in a forward stepwise sequence, exploring how different combinations of variables fit and perform in the regression model. Using criteria that assessed p values significance, multicollinearity measured by the variance inflation factor (VIF), normality of residuals and clustering of residuals in space (table 2), we selected the variables that best explained the outcome and met the criteria of a well-specified regression model and explored these through a more rigorous OLS modelling exercise.

### Global regression model
The performance of the OLS models chosen from the exploratory regression were assessed based on the magnitude of the adjusted $R^2$ values. In addition, we checked for significance of p values for the model coefficients. Multicollinearity between different variables in a model was checked using the lower VIF threshold of five. The Koenker statistic (p<0.01) was used to check if the relationships being modelled were consistent (either due to non-stationarity or heteroskadisticity), while the Wald statistic was used to assess overall model significance. The Jarque Bera test (p<0.01) was used to check if model predictions were biased (ie, if the model residuals were normally distributed). The model that performed best and met these criteria was selected for further analysis to create a locally specified model.

### Local regression model
The geographically weighted regression (GWR) technique was used to develop a second model, which extended the output from OLS, to explore spatial non-stationarity of effect of the variables. This allowed for the new model to account for spatial structure in estimating local rather than global model parameters.[34 35] We foresee this to be an important step to creating interventions that are locally specific and an important part of more precisely targeting

interventions. As part of the modelling process, the spatial weights based on the geographic proximity of observation are applied to give more weight to values that were closer together. GWR4 software[36] was used for this part of the project. The geographic variability test was conducted to assess if there was significant non-stationarity in the coefficients after applying GWR. This test compares the geographically varying parameters with those in the fixed global model, where a negative difference (abbreviated 'DIFF OF CRITERION' in GWR4), indicates significant variation in parameter estimates across space.[36] We also assessed the performance of the GWR model using the newly calculated values of the adjusted $R^2$.

### Patient and public involvement
Previous research[24] that elicited community perceptions of risk factors related to adverse maternal outcomes from women, their male counterparts and community leaders in the study area informed the choice for some of the variables. This study used FGDs and in-depth interviews and has been published through other avenues. The same results have also been disseminated through an outreach workshop to the ministry of health in Mozambique. Further research that has received recent funding will communicate the identified risk factors through the use of a mobile app.

## RESULTS
### Community perspectives and choice of variables
The full list of community-level variables that were considered is presented in table 1. These variables, gathered from the results of focus group discussions, semistructured interviews and the Delphi consensus, represent the local perspectives, expert views and a priori study knowledge. The community perspectives that informed some of our choice variables are reported in full separately.[19 24]

### Statistical analysis
#### Descriptive statistics
The geographic pattern for the rates of the combined adverse maternal and perinatal outcome is shown in figure 2, where the clusters of Ilha Josina+Calanga, Mazivila and Chissano had the localities with the highest rates of the combined outcome.

For women with completed pregnancies, community-level scores for census and geographical variables are summarised in table 3. The average age of women with complete pregnancies was 26. Ninety-one per cent of households reported that the head of household lived in the household. There was large variability in the percentage of households with an improved water source, with the lowest locality level score being 11.9%, while the highest was 99%. The number of households with an improved toilet facility was very low, with the best locality level score for this variable being 31.6%. An average of 5.6% of all households reported owning a private vehicle. Most women reported being in a marital union (70.9%).

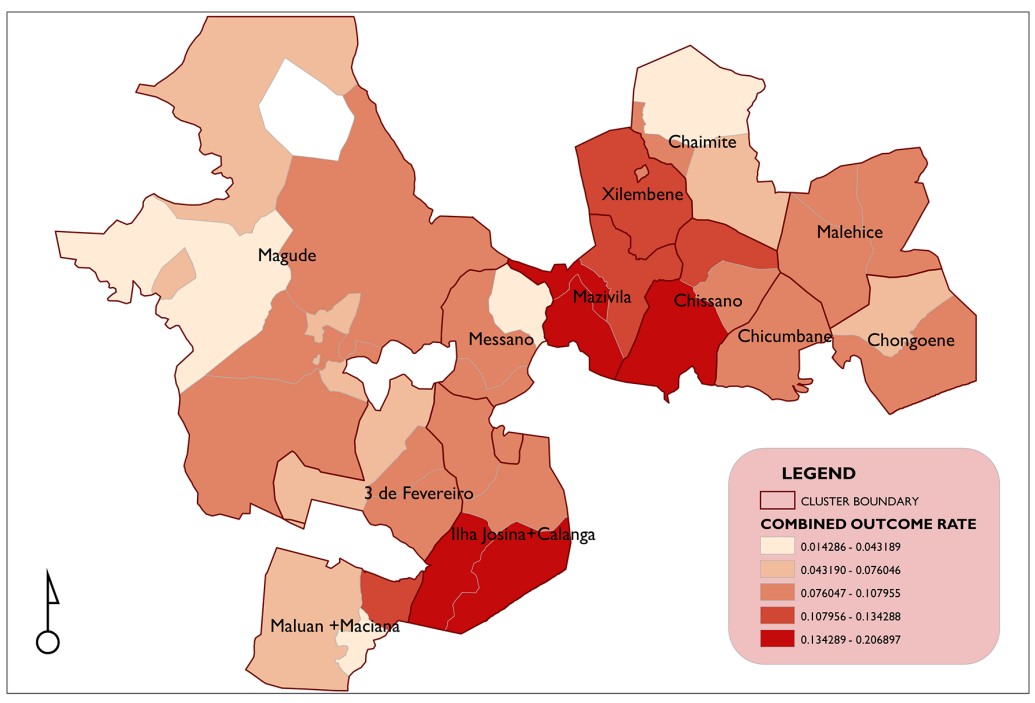

**RATES FOR THE COMBINED ADVERSE OUTCOME**

Maternal, fetal and neonatal deaths

**Figure 2**  Geographic pattern for the rates of the combined adverse outcomes.

| Table 3 Summary statistics | | | | |
|---|---|---|---|---|
| **Community level variable** | **Min** | **Max** | **Mean** | **SD** |
| Census variables | | | | |
| 1. Average age of reproductive age women (years) | 24.30 | 28.00 | 26.36 | 0.90 |
| 2. Household head's education (years) | 3.60 | 7.30 | 5.47 | 0.88 |
| 3. Household head's availability (proportion) | 0.80 | 1.00 | 0.91 | 0.07 |
| 4. Water source score (%) | 11.90 | 99.00 | 54.68 | 25.97 |
| 5. Latrine score (%) | 0.00 | 31.60 | 15.48 | 8.35 |
| 6. Private transportation score (%) | 0.00 | 12.30 | 5.57 | 3.05 |
| 7. Reproductive age women's education (years) | 3.80 | 7.20 | 5.29 | 0.95 |
| 8. Fertility rate (no. per woman) | 2.40 | 3.80 | 2.89 | 0.29 |
| 9. Reproductive age women's marital status score (%) | 52.40 | 88.90 | 70.85 | 8.47 |
| 10. Reproductive age women's unemployment rate (proportion) | 0.00 | 0.40 | 0.11 | 0.10 |
| 11. Family support (%) | 61.30 | 100.00 | 85.61 | 10.46 |
| 12. Community support (%) | 0.00 | 17.30 | 3.59 | 3.86 |
| 13. Financial autonomy in pregnancy (%) | 0.00 | 45.90 | 22.15 | 8.43 |
| Geospatial variables | | | | |
| 14. Access to primary health facilities (hours) | 0.18 | 1.75 | 0.61 | 0.36 |
| 15. Access to secondary health facilities (hours) | 0.29 | 2.80 | 1.20 | 0.54 |
| 16. Access to tertiary health facilities (hours) | 0.97 | 4.32 | 2.08 | 0.74 |
| 17. Isolation (travel time to nearest main road in hours) | 0.10 | 2.01 | 0.54 | 0.42 |
| 18. Flood proneness (%) | 5.61 | 8.27 | 6.75 | 0.67 |
| Rate of adverse outcomes per live birth (maternal+neonatal+ miscarriages+stillbirths) | **0.01** | **0.21** | **0.10** | **0.05** |

**Table 4** OLS model

| Variable | Coefficient | SE | t-Statistic | VIF |
|---|---|---|---|---|
| Intercept | −0.194483 | 0.19802 | −0.98214 | – |
| Isolation (hours) | 0.033353* | 0.01464 | 2.27805 | 2.06598 |
| Flood proneness (%) | 0.023936** | 0.00754 | 3.17587 | 1.35311 |
| Latrine score (%) | −0.003094*** | 0.00072 | −4.28457 | 1.92347 |
| Family support (%) | −0.001274* | 0.00051 | −2.48055 | 1.52648 |
| Age of reproductive age woman (years) | 0.034653*** | 0.00826 | 4.19544 | 2.93335 |
| Fertility rate (no. of children) | −0.222607*** | 0.02942 | −7.56575 | 3.90853 |

*P≤0.05; **p≤0.01; ***p≤0.001.
Multiple $R^2$=0.75; adj $R^2$=0.69.
OLS, ordinary least squares; VIF, variance inflation factor.

The reported rates of unemployment were surprisingly low (mean 11%), given how this was perceived as an important risk factors in the focus group discussions. The proportion of women who indicated that they would receive either financial, transport or emotional help from family or a neighbour in the event of a pregnancy-related need was 85.6%, while only 3.6% would receive the same from community groups. Twenty-two per cent of households indicated that the woman was empowered to make financial decisions concerning her pregnancy.

The average travel time to the primary health facilities was 0.6 hours using public transport. For women who were referred to secondary facilities, this was calculated to take an average of 1.2 hours, assuming that they used public transport to primary facilities and an ambulance to secondary facilities. For tertiary facilities, it was calculated to take and average of 2.1 hours using the same combination of transport modes. The most isolated communities required women to walk 2.0 hours to the nearest main road, while the closest were less than 6 min (0.1 hours) away. The ease of travel through communities reduced by an average of 6.8% because of flooding and precipitation during the 12 months prior to the household census.

### Global model
Through exploratory regression, we identified six variables that met prespecified criteria for inclusion in the model (online supplementary file 1). The resulting OLS model is illustrated in table 4. The adjusted $R^2$ was 0.69 (online supplementary file 2). A full record of the diagnosis for the OLS is provided as online supplementary material. The graduated colour classification maps describing the magnitude of the variables across the study area is presented in figure 3.

The OLS model shows that as the degree of isolation increases, there is an effect of increasing rates of adverse outcomes (p≤0.05). There is significantly more isolation in the western region of the study area, particularly for the Magude, Ilha Josina and Calanga clusters, making women in these areas more vulnerable to the effect of isolation.

Communities that were shown to have more fragile and flood prone road infrastructure during the year prior to data collection were also shown to have elevated rates of the adverse outcome (p≤0.01). Regions in the Ilha Josina+Calanga, Mazivila and Chaimite clusters were the most affected by flooding and precipitation.

Higher rates of availability of an improved latrine are associated with lower rates of the adverse outcome (p≤0.001). Regions in the Ilha Josina+Calanga, Magude and Chaimite clusters had the lowest rates of improved latrines, while 3 de Fevereiro and Chongoene had the highest rates.

Family support emerged as an important characteristic in reducing the rates of adverse outcomes (p≤0.05). Although the levels of family support are relatively high for all localities (mean 85.6%), there is a north–south divide between the communities that have higher and lower rates of family support with Maluana+Maciana, Ilha Josina+Calanga and 3 de Fevereiro having the highest rates while Mazivila, Chissano and the northern region of Magude have lower rates.

Average age of women with a completed pregnancy was positively associated with rates of the adverse outcome (p≤0.001), while fertility rates are negatively associated with the rates of adverse outcomes (p≤0.001). A closer look at the age aggregated data for age and fertility rates (figure 4) reveals a bidirectional trend in the both relationships with the combined outcome. Rates of the combined outcome are relatively higher for the younger ages of 12 years old and decrease until age 20 years, where they begin to rise until age 49 years. A similar pattern exists for fertility rates with women who have lowest fertility rates experiencing higher rates of adverse outcomes that decline until fertility rates of approximately 2.2 before rising again for higher fertility rates.

### Local model
The GWR model indicated that the effect on the combined outcome was geographically non-stationary for three of the six variables (isolation, latrine score and family support)

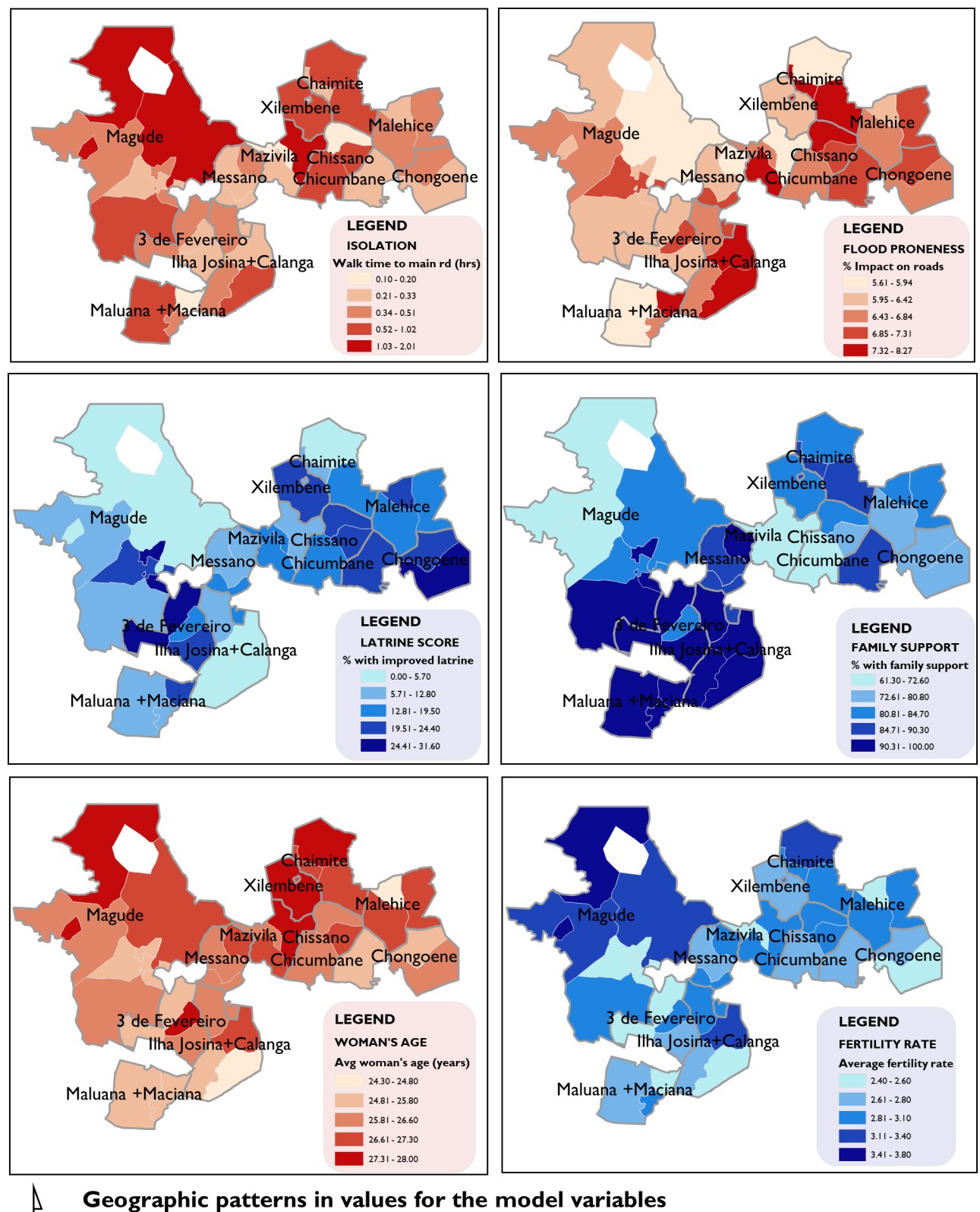

**Geographic patterns in values for the model variables**

Blue is for protective factors and red for risk factors

**Figure 3** Geographic patterns in values for the model variables.

as indicated by the negative DIFF OF CRITERION values in table 5 (online supplementary file 3). There was no non-stationarity for the effect of flood proneness, age of the woman and fertility rate. The performance of the local model improved modestly from the global model to an adjusted R² value of 0.71, explaining a further 2% of the variability in the outcome. A graduated colour classification was used to describe the magnitude of the effect in each of

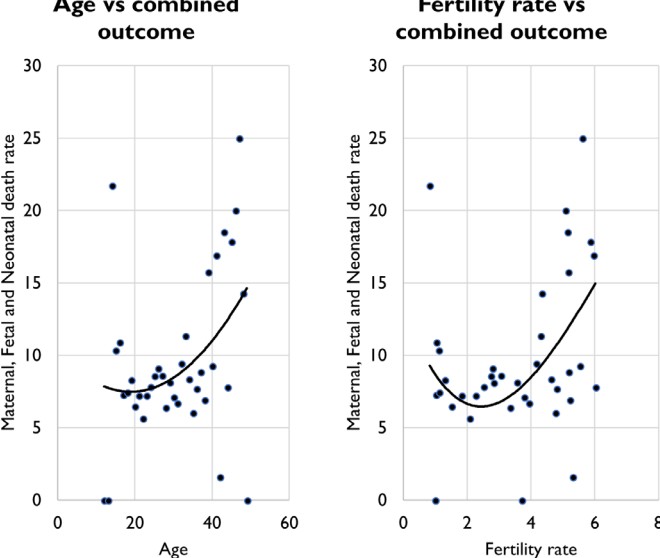

**Figure 4** Age and fertility rates compared with adverse outcomes. The classic J shape graph for both.

the localities. The transcripts of the GWR results are available as supplementary material.

The general direction of the effect in the global model is preserved in the local model (figure 5). Isolation has a general effect of increasing risk for adverse outcomes across the entire study area. However, the effect is more pronounced by a factor of approximately 53% in the western region of Magude compared with the east in Chongoene. A similar pattern is observed for the effect of flood proneness on the combined outcome, with the western regions of the study area being at greater risk, though the magnitude of the differential effect is much smaller (8%). This is also even though the same areas in the west had lower level of proneness to floods (figure 3).

Proportion of household with an improved latrine was associated with decreasing the adverse outcomes for all regions under study, although this effect is greatest in magnitude for the eastern region of the study area by a factor of about 12%. Family support has a greater effect of reducing the rates of adverse outcomes in the eastern regions by a factor of about 45%. Average woman's age had an effect of increasing the outcome by up to 15% more in the west than the east. The variability in the association of fertility rates to the rates of adverse outcomes

**Table 5** Results of the geographic variability test

| Variable | Diff of criterion |
|---|---|
| Intercept | 0.176197 |
| Isolation | −1.224025 |
| Flood proneness | 0.530483 |
| Latrine score | −0.156581 |
| Family support | −2.551618 |
| Age of reproductive age woman | 0.073263 |
| Fertility rate | 0.190761 |

was also non-stationary with the highest effect 5% more than the lowest.

## DISCUSSION

This study has explored the place-specific factors that are associated with rates of adverse maternal and perinatal outcomes. Information gathered through FGDs and semistructured interviews enabled us to measure the context-specific determinants that were thought to be related to adverse maternal and perinatal outcomes. Some of the variables from the FGDs and interviews were indeed significantly associated with the rates of our combined outcome and include family support, geographic isolation and access to an improved latrine facility. Other noteworthy variables include flood proneness, average age of reproductive age women with completed pregnancy and fertility rates. The effect and the magnitude of the effect of these determinants of the outcome varied between communities though the direction of the effect was largely constant.

Fertility rates were the only variable where the direction of the effect is contrary to common expectation. However, fertility rate was the most significant variable both in the global model (p≤0.001) and as a single variable in the exploratory regression, so the observed pattern is unlikely a result of effect modification. Instead, it is possible that there are other pervasive factors at play. The graph for fertility rates plotted against the rates of adverse outcomes (figure 4) indicated that adolescent women with completed pregnancies also have lower fertility rates but tend have high rates of the adverse outcome.[26 37] This trend could have possibly skewed the direction of the effect of this variable. This pattern would be possible as the most completed pregnancies (61%) were from women with fertility rates less than 2.2; therefore, community-level averages would naturally carry more weight from these women's records.

This is the first time that the place-specific sociocultural and environmental factors related to adverse maternal outcomes have been explored in this region of Mozambique. Similar methods have been used in the USA.[38] However, most of these studies emphasise health systems-related variables and how they relate to adverse maternal outcomes. This project's approach of going into communities to meet with local stakeholders is aligned with upcoming strategies within the SDGs for improving maternal health (United Nations, 2015). Core to these new global health strategies is an emphasis on multisectoral interventions that broadly consult multiple local stakeholders to understand the context-specific factors that may be related to population level health trends. The value of geographical techniques to these new strategies is demonstrated in two ways in this project.

First, we used Geographical Information Systems (GIS) to design new indicators for some of the context-specific variables perceived to be related to adverse maternal outcomes. Our measure of isolation for example (distance

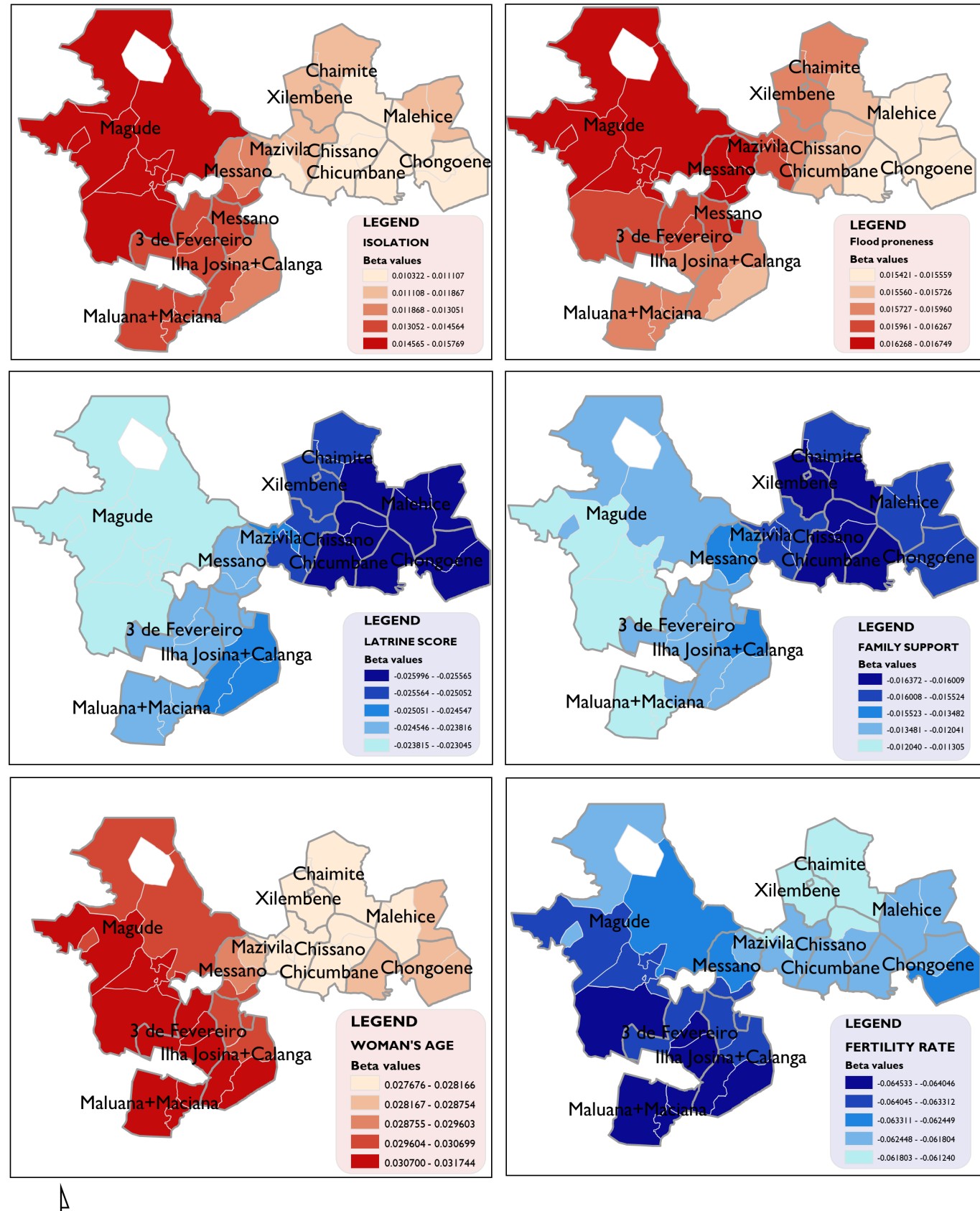

**Geographic variation in the effect of the variables on the combined outcome**

Darkening shade of blue indicates increasing protective effect while darkening shade of red indicates increasing risk effect.

**Figure 5** Geographic variation of beta coefficients.

from highway) was designed based on local knowledge from the FGDs and interviews and implemented in a GIS. Other GIS variables on access to care were generated based on an understanding from the FGDs and relevant literature that most women would either walk or use public transport to primary facilities and be driven to higher level facilities.[39–41]

Second, the use of geographically explicit techniques, such as GWR, enhances the ability to elucidate the spatial structure of the determinants of health and is in line with a drive within global health (expressed through the SDGs) to measure more granular subnational trends in health.[2] Evidence from the local regression model developed for this project highlights that different determinants matter to different extents in different places. This understanding is valuable, particularly for designing and targeting population-level interventions for improving maternal and child health because it allows for action in specific places to be informed by what is known to elevate risk the most. This is critical for achieving impact and improving health in a resource limited settings like Mozambique.[42 43]

A key limitation for this study is that the results can only be applied at the population level and should not be used to predict outcomes at the individual woman level. This phenomenon is termed ecological fallacy[44] and is a key drawback for conducting population-level studies like this one. Furthermore, this project created a combined outcome for maternal, fetal and neonatal outcomes. The implication for this is that the results may not mirror the actual associations with any of the three outcomes in the combined outcome if considered separately. However, the observed patterns address more broadly, an approach to improving maternal health along the continuum of care that includes thinking about fetal, newborn and child survival together.[5] Furthermore, the combined outcome better captures the true impact on families, as tragedies of death of mothers, fetuses and infants do not happen in isolation.

## CONCLUSIONS

The geographic perspective contributes to new strategies for improving global maternal and perinatal health by providing the tools required to understand local contexts and associated determinants. It also helps to elucidate the geographic structure of associations between these determinants and adverse maternal and perinatal health outcomes. This is crucially important for targeting interventions and can help to operationalise some of the key strategies within the new SDG. While the patterns that characterise the findings of this project are specific to the region of Southern Mozambique and may not be transferable to other settings, this research design could certainly be translated to help with understanding the local factors that elevate risk for adverse maternal and perinatal outcomes. It was outside the purview of this research to explore how this evidence could be translated into action

on these community-specific social determinants, and future work should address this knowledge gap .

## Author affiliations
[1]Surveying and Geomatics Department, Midlands State University Faculty of Science and Technology, Gweru, Midlands, Zimbabwe
[2]Department of Geography, Simon Fraser University, Vancouver, British Columbia, Canada
[3]Department of Obstetrics & Gynaecology, University of British Columbia, Vancouver, British Columbia, Canada
[4]Centro de Investigacao em Saude de Manhica, Manhica, Maputo, Mozambique
[5]Department of Epidemiology, Biostatistics and Occupational Health, McGill University, Montreal, Quebec, Canada
[6]Department of Obstetrics and Gynaecology, Kings College London, London, London, UK
[7]Faculty of Medicine, Universidade Eduardo Mondlane, Maputo, Mozambique
[8]Department of Medicine, Yale New Haven Health System, New Haven, Connecticut, USA

**Twitter** @ptmakanga, @placelaertlabs

**Collaborators** CLIP Working Group: Eusébio Macete, Anifa Vala, Felizarda Amose, Rosa Pires, Zefanias Nhamirre, Marta Macamo, Rogério Chiaú, Analisa Matavele, Ariel Nhancolo, Silvestre Cutana, Ernesto Mandlate, Salésio Macuacua, Cassimo Bique, Sibone Mocumbi, Emília Gonçálves, Sónia Maculuve, Ana Ilda Biz, Dulce Mulungo, Orvalho Augusto, Paulo Filimone, Vivalde Nobela, Corsino Tchavana, Cláudio Nkumbula, Jeffrey Bone, Dustin Dunsmuir, Sharla K Drebit, Chirag Kariya, Mai-Lei Woo Kinshella, Jing Li, Mansun Lui, Beth A. Payne, Asif R Khowaja, Diane Sawchuck, Sumedha Sharma, Domena K. Tu, Ugochi V. Ukah.

**Contributors** PTM, CS, NS, PvD and TF made substantial contributions to conception and design of the project. FCV, CS, KM, HB and ES led the acquisition of in country data. PTM, UVU, NS and TL led the analysis and were involved together with all the authors in the interpretation of the results. PTM, LAM, TF, NS and MV made substantive contributions to writing the first complete version of the article, and all the authors were involved in revising it critically for important intellectual content. All authors approve of the final version of the paper.

**Funding** This work was part funded by Grand Challenges Canada – Stars in Global Health programme (Grant 0197) and was conducted as part of the Pre-eclampsia/Eclampsia, Monitoring, Prevention and Treatment (PRE-EMPT) initiative supported by the Bill & Melinda Gates Foundation.

**Competing interests** None declared.

**Patient consent for publication** Not required.

**Ethics approval** Ethics approval for the study was obtained by the research and ethics boards at University of British Columbia, Simon Fraser University and Centro de Investigação em Saúde de Manhiça.

**Provenance and peer review** Not commissioned; externally peer reviewed.

**Data sharing statement** All data that have been used in this research will be available upon request after the main papers from the Community-Level Interventions in Pre-Eclampsia (CLIP) trials have been published.

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
