## [Reviewer comments · BMJ Open]

This paper was submitted to a another journal from BMJ but declined for publication following peer review. The authors addressed the reviewers' comments and submitted the revised paper to BMJ Open. The paper was subsequently accepted for publication at BMJ Open.

(This paper received three reviews from its previous journal but only two reviewers agreed to published their review.)

ARTICLE DETAILS

TITLE (PROVISIONAL)	The place-specific factors associated with adverse maternal, and perinatal outcomes in southern Mozambique: A retrospective cohort study
AUTHORS	Makanga, Prestige Tatenda; Sacoor, Charfudin; Schuurman, Nadine; Lee, Tang; Vilanculo, Faustino Carlos; Munguambe, Khatia; Boene, Helena; Ukah, UV; Vidler, Marianne; Magee, Laura; Sevene, Esperanca; von Dadelszen, Peter; Firoz, Tabassum

VERSION 1 – REVIEW

REVIEWER	Di Gennaro Francesco Department of Infectious Diseases Bari, Italy
REVIEW RETURNED	26-Jun-2018

GENERAL COMMENTS	For my opinion this is a well written and well Thought article. I very appreciate the paper. Better understand the specific factors of adverse maternal and perinatal outcomes play a central role of efficacy for public health measures I only suggest two interesting paper on this issue in Mozambique -Marotta C et al. Pathways of care for HIV infected children in Beira, Mozambique: pre-post intervention study to assess impact of task shifting.BMC Public Health. 2018 Jun 7;18(1):703 - Marotta C et al. The At Risk Child Clinic (ARCC): 3 Years of Health Activities in Support of the Most Vulnerable Children in Beira, Mozambique. Int. J. Environ. Res. Public Health 2018, 15
---

REVIEWER	Saifuddin Ahmed Bill and Melinda Gates Institute for Population and Reproductive Health, USA
REVIEW RETURNED	02-Jul-2018

GENERAL COMMENTS	The place-specific factors associated with adverse maternal, and perinatal outcomes in southern Mozambique This paper examines individual, familial and contextual factors that are associated with maternal, fetal and neonatal mortality in southern Mozambique. Although the study addresses an important knowledge gap, the paper needs substantial improvement in organizations and presentations. I also have a number of
---

questions about the analytical methods. I summarize my major comments below:

1. What is the sample size of the regression analyses? It is difficult to ascertain the sample size of the analyses from the texts in the paper: 9172 women? 8796 births? 12 clusters? 36 smaller regions? From the description, I assume that the analyses were performed at the cluster level. If the unit of analysis is the clusters, the study may not have adequately utilized the large individual level sample size of women. Information on the sample size is needed to understand the validity of the analyses in the paper.

2. Similarly, it is difficult to understand the process of creating "outcome scores." I understand that the numerators of the scores are maternal, fetal and neonatal deaths. Fetal deaths include both spontaneous abortions and still births, I presume. The underlying causes (risk factors) of spontaneous abortions are quite different than from the causes of neonatal deaths. Importantly, I have difficulty to understand the denominator of the score calculation: women or pregnancies or both? Because of twin pregnancies/births, the number of women is usually less than the number of pregnancies. Of the twins, one may have survived and one have died (twins are often excluded from neonatal mortality determinant studies to avoid biological heterogeneity in study sample). The authors need to detail the method of outcome score creation.

3. Based on three outcome binary variables (with extremely high zero values), the distribution of such a score is likely to be extremely skewed. OLS may not be a good choice for highly non-normal data.

4. I suggest that the authors use the Poisson or negative binomial model (in case of over-dispersion suspected) with the "counts of mortality events" as the outcome and the (log) number women/pregnancies as offset. This analysis will utilize the information of the individual level large sample size of the study. The analysis also needs to take into account clustering nature of data. It is possible to fit GWR with glm specification that allows "poisson" as family name.

5. It is increasingly recognized that stepwise or similar approach to fit parsimonious model is problematic: it increases Type-I error rates and R-square vales. In addition, it hides negative results. Authors may consider fitting the model based on their conceptual framework of relationships between the covariates and outcomes; there are well known conceptual frameworks for maternal and neonatal mortality in the field. Theoretical model fitting is preferred over models with data-driven selection criterion.

6. The description of the rationales for using the geographically weighted regression is not clear:

"The Geographically Weighted Regression (GWR) technique was used to develop a second model, which extended the output from OLS, accounting for spatial structure to estimate local rather than global model parameters (32,33). As part of the modelling process, the spatial

weights based on the geographic proximity of observation are applied to give more weight values that were closer together."

My understanding is that the paper is examining spatial non-stationary relationship/effect of the variables with the GWR model and that should be directly stated and explain the purposes of examining non-stationarity for the general readers.

	7. I suggest that the authors avoid explaining the coefficients if the meaning is not clear or interpretable (“1 hour increase in walking time to the nearest main road results in a 3.3% increase in the rates of the combined outcome”: what is the meaning of 3.3% in terms of mortality (mother/fetal/neonatal)? Explain if this could be converted to XX deaths per 100,000 XX). 8. Lack of a conceptual framework is a concern to me for the selection of variables in the study. As an example, the study found that improved latrine is a significant variable. However, a latrine may reflect underlying economic status or hygienic practices that may increase the risk of postpartum sepsis and neonatal sepsis. What is the implication of this finding? In this study, which uses a parsimonious model, toilet variable may reflect underlying SES condition and urban/rural residence status of the sample. 9. I suggest that the paper limits only to “quantitative analyses” and the authors consider presenting the qualitative study results in a separate paper. Qualitative study methods or results have no impact or implications on the analysis decisions or study inferences. The selected variables are well known and established in the maternal and neonatal mortality studies.
--	--

VERSION 1 – AUTHOR RESPONSE

Response to Reviewer 1

We would like to thank the reviewer for the kind gesture and suggestions for references. However, we decided not to include the suggested references as their focus was child health and HIV, which deviates from the core health matters that are addressed in the present manuscript.

Response to Reviewer 2

Comment 1. Geographic unit of analysis

The geographic unit of analysis was the locality (36 smaller regions). Our ethics did not permit for individual level geographical analysis hence the need to aggregate the data and calculate community level estimates. The text that captures this response in the paper now reads as follows:

These variables and other community-level estimates for the variables captured in the census were calculated for each locality in the study area as described in Table 1. Both the census and geographical data were aggregated into community-level averages for each of the chosen variables, as ethical approval did not allow to analyse the location data at the level of the individual woman.

Comment 2 – The combined outcome

Spontaneous abortions were not included in the outcome. We included maternal deaths, miscarriages, still births and neonatal outcomes. We provide references to justify the choice for the combined outcomes. The denominator was live births for the women with completed pregnancies. We believe that this accounted for the potential problem with twin birth outcomes. A statement to this effect has been included in the statistical methods and variables section.

Comment 3 and 4 – Appropriateness of methods

In response to comments 3 and 4: We conducted several other tests to ensure that the resulting model satisfied the conditions required for a well specified OLS model. We checked for significance of p values for the model coefficients. Multi-collinearity between different variables in a model was checked using a Variance Inflation Factor (VIF) threshold of five. The Koenker statistic ($p < 0.01$) was used to check if the relationships being modelled were consistent (either due to non-stationarity or heteroskadisticity), while the Wald statistic was

used to assess overall model significance. More importantly, the Jarque Bera test ($p < 0.01$) was used to check if model predictions were biased (i.e. if the model residuals were normally distributed).

With regards to Zero heaviness. Our aggregated outcomes was not zero heavy as all records were collapsed to outcome rates for 36 admin units.

Comment 5 – Model fitting

We did not fit the model using an exclusively mechanical process of variable selection. Our starting point was considerations from FGDs and IDIs from a component of our study that has been published separately. We also went through a depi consensus to isolate variables that matter based on expert opinion. After the screening we think that all the variables that we had prioritized were valuable, and only then did we perform the exploratory process to identify the variable that best told the story of the trends in the outcome.

Comment 6 – Geographically weighted regression

We have added the following text to indicate the value that GWR brings to the results. Text now reads;

The Geographically Weighted Regression (GWR) technique was used to develop a second model, which extended the output from OLS, to explore spatial non-stationarity of effect of the variables. This allowed for the new model to account for spatial structure in estimate local rather than global model parameters. We foresee this to be an important step to creating interventions that are locally specific, and an important part of more precisely targeting interventions.

Comment 7 and 8 – Explaining coefficients

In response to comments 7 and 8, I somewhat agree with the reviewer that some of the variables to be symptomatic of a bigger problem rooted in low SES. However, the specific example of latrines could also be linked to poor hygiene's link to infections. We do not have the data to explore evidence for these deep-rooted linkages, and I have acknowledged this as a limitation. We have also removed many of the statements that explained the magnitude of the effect against the outcome as we were working with a combined outcome and cannot single out predicted deaths rates for each of the constituent outcomes.

Comment 9 – Limiting paper to quantitative analysis

We agree the with author that the qualitative components were thin and have decided to take them out of this paper as they have been reported in a separate publication. We now simply acknowledge the qualitative portion of the broader study as having informed some of our choice variables. However, we do not describe these variables and how they emerged in the community as they have been communicated through the other publication that we reference in the present manuscript.